# Psychosocial Risk Prevention in a Global Occupational Health Perspective. A Descriptive Analysis

**DOI:** 10.3390/ijerph16142470

**Published:** 2019-07-11

**Authors:** Francesco Chirico, Tarja Heponiemi, Milena Pavlova, Salvatore Zaffina, Nicola Magnavita

**Affiliations:** 1Health Service Department, State Police, Ministry of Interior, Milan 20162, Italy; 2Post-graduate School of Occupational Medicine, Università Cattolica del Sacro Cuore, Rome 00168, Italy; 3Social and Health Systems Research Unit, National Institute for Health and Welfare, Helsinki FI-00271, Finland; 4Health Services Research (HSR), CAPHRI, Maastricht University Medical Center, Faculty of Health, Medicine and Life Sciences, Maastricht University, Maastricht 6202 AZ, The Netherlands; 5Occupational Health Unit, Ospedale Pediatrico Bambino Gesù, Rome 00165, Italy; 6Department of Woman/Child and Public Health, Fondazione Policlinico “A. Gemelli” IRCCS, Rome 00168, Italy

**Keywords:** global health, health inequalities, legislation, mental health, psychosocial hazard, public health policy, workplace violence, job strain, psychosocial hazard, occupational health

## Abstract

This study aimed to find out which countries around the world require psychosocial hazards and workplace violence to be assessed by employers through a mandatory occupational risk assessment process and to compare the type of legislation between countries. We systematically searched the International Labour Office (ILO) “LEGOSH” database for documents published during the period between December 2017 and February 2018. The search included 132 countries, of which 23 were considered as developed and 109 as developing according to the United Nations. Our review showed that most countries (85, i.e., 64%) have not included mandatory psychosocial risk assessment and prevention in their national occupational safety and health legislation. Moreover, we found differences between developed and developing countries, showing that developed countries more frequently have legislative measures. Within developed countries, we also found differences between countries following the Scandinavian model of workplace health and safety culture and other countries. Moreover, in many countries, workplace violence was prohibited only if it involves an offence to moral or religious customs. In conclusion, the marked difference in psychosocial hazards and workplace violence regulations among countries leads to unequal levels of workers’ protection, with adverse effects on global health.

## 1. Introduction

The existence of occupational diseases was first recognized nearly three centuries ago when Bernardino Ramazzini published “De Morbis Artificum Diatriba” [1]. Despite this, the global burden of occupational diseases is continuously growing around the world [2], and this requires a growing commitment on the part of all nations.

During the last century, many countries across the world have established laws for the prevention of occupational safety and health (OSH) risks. Generally, these legislative measures have especially taken into account traditional (chemical, physical or biological) risk factors. However, OSH legislation less frequently takes into consideration the so-called “fourth group”, that is the psychosocial occupational risk factors. Psychosocial hazards (PSH) have been identified as one of the key emerging risks in OSH [3]. They are defined as “those aspects of work design and the organization and management of work, and their social and environmental context, which may have the potential to cause psychological or physical harm” [4]. They originate from “interactions between and among work environment, job content, work organization and workers’ capacities, needs, culture, personal extra-job considerations that may, through perceptions and experience, influence health, work performance and job satisfaction” [5]. Such a broad definition includes countless possible stressors. Occupational or job-related stress is just one of them.

Despite its frequent use, no agreement upon the definition of job stress currently exists; it, however, refers to distinct job stressors, or stimuli in the workplace, leading to psychological strains or negative psychophysiological responses or reactions [6]. According to the WHO (World Health Organization), this negative response occurs when work demands and pressures do not match employees’ abilities and instead challenge employees’ ability to cope [7]. Job stress can be caused by poor work organization and work design, poor management, unsatisfactory working conditions, and lack of support from colleagues and supervisors [8]. Workplace violence (WV) can be included among the harmful psychosocial stress factors, both because sometimes it originates from inside of the work environment (assuming the character of bullying or mobbing), and because the attacked people often develop a condition of distress, which in turn makes them prone to violence, in a cyclical relationship [9,10]. According to the Occupational Safety and Health Administration (OSHA) (2002), WV can be defined as any “violence or the threat of violence against workers, which can range from threats and verbal abuse to physical assaults and homicide” [11].

The importance of PSH emerges from studies conducted in workplaces in many parts of the world. In Europe, where regional figures are available, stress/strain is the second most frequently reported work-related health problem. A European survey (EU-OSHA, 2009) showed that 50–60% of all lost working days are attributed to job strain, and the number of people suffering from stress-related conditions caused or made worse by work is likely to increase [12]. In the Asia-Pacific Region, one in five Australian employees reported to be mentally unwell in the past 12 months, and it was estimated that untreated mental health conditions cost Australian workplaces approximately $10.9 billion per year [13]. In Japan, a national report showed that almost every third worker reported suffering from job strain-related anxiety disorders during the previous year [14]. In the American nations, cross-sectional studies showed that more than 10% of respondents to the First Central American Survey on Working Conditions and Health (2012) reported various job strain-related symptoms, ranging from feeling depressed to sleep problems [15]. In the USA, more than 570,000 workers a year experience non-fatal WV [16,17]. Conversely, almost no information about prevalence or incidence rates of PSH- or WV-related diseases has been identified in African and other developing countries [14].

Due to their importance for health, safety and productivity, PSH and WV must be addressed and prevented in the workplace. However, the employer’s decision to establish an Occupational Health and Safety (OHS) service, including measures for health surveillance of workers, health promotion, counselling and disability management, can only be established on the basis of a double legitimacy, being based on both scientific evidence (i.e., scientific legitimacy) and laws (i.e., legal legitimacy) [18]. Regulation of these topics is of importance, as it is well known that greater quality of healthcare systems’ prevention efforts is linked with lower occupational fatality rates and higher levels of workers’ health [19]. Reviewing the OSH regulatory framework is of great importance because the presence of differences in regulatory provisions could correspond to differences in the levels of safety and health in the workplace. By making these differences explicit in our review, we provide a base for further policy discussion and improvements.

Not all countries in the world utilize the same level of occupational risk prevention. The cultural and social differences that underlie the different regulatory situations are also the basis of different development models. In recent years, the process of globalization and liberalization have often increased health inequities [20]. For this reason, it is relevant to know whether or not the presence of laws or regulations in the countries around the world really promotes the development of effective OSH services for the prevention of PSH.

In this review, we aimed to evaluate the OSH regulatory frameworks developed by the countries worldwide, with a special focus on the differences between developing and developed countries and within developed countries, which have more resources to achieve occupational risk prevention. More specifically, our study aimed to clarify whether countries around the world require PSH and WV to be assessed by employers through the mandatory occupational risk assessment process. Such a legal requirement could inform policymakers and key stakeholders for improving occupational health and safety policy on PSH and WV in the workplace.

## 2. Materials and Methods

The International Labour Office (ILO) Global Database on Occupational Safety and Health Legislation (LEGOSH) is a database that compiles legislation in the OSH field [21]. The continuous updating of the collection of laws makes it one of the most suitable tools for evaluating the state of the main elements of OSH legislation, including OSH management and administration, employers’ duties and obligations, workers’ rights and duties, OSH inspection and enforcement, among others. We have systematically consulted the database, during the period between December 2017 and February 2018, to obtain information on the legislative provisions aimed at preventing PSH in different countries and to compare, from a global perspective, the different situations.

The LEGOSH classification structure is based on a comprehensive set of 11 themes, which follow and capture the main part of the key ILO standards. LEGOSH sub-divides the category of PSH into two sub-groups: (a) “psychosocial risks” and (b) “occupational violence”. LEGOSH has a user-friendly interface, which allows, but is not limited to: a) access to the synthesis of OSH legislation in English; b) comparison of the legislation of several countries or regions on a particular subject (by using the function “Compare countries”); and (c) conducting customized searches. LEGOSH was independently screened and assessed by the authors of this study. We specifically analyzed the Subtheme 9.5 (“Psychosocial hazards”) of Theme 9 (“Specific hazards or risks”), which provides an overview of the main pieces of legislation covering specific hazards or risks, including psychosocial ones.

Our search was limited to the 132 countries included in the LEGOSH database. Our search was firstly carried out for each country by using the function “Search”. Then, we checked for each country (“Africa all”, “Americas all”, “Arab States all”, “Asia all” and “Europe all”) by using the function “Compare countries”. Finally, we analyzed our findings drawn from all OSH legislation by comparing countries from developed continents (Europe, North America, Oceania) with countries from developing continents (Asia, Africa, Central and South America).

Criteria to include each country in “developed” or “developing” countries were based on the 2018 World Economic Situation Prospects report by the United Nations (United Nations, 2018), which arranges countries around the world into three classes: (a) developed economies, (b) economies in transition and (c) developing countries [22]. Economies in transition and developing countries were considered as one group in our study. In this study, we equated “no data available” about the mandatory assessment of PSH and WV with “no regulation”. Moreover, we did not consider criminal law or laws against sexual violence that may be present in the codes of most countries but are not specifically referred to in the workplace.

We firstly performed a directed qualitative content analysis based on predefined themes (thematic analysis on PSH and WV), followed by a synthesis of the results and a narrative description, which is illustrated with tables.

## 3. Results

The characteristics of the legislation of the various countries are reported in Appendix A and Appendix B. The general characteristics of the OSH legislation reviewed are presented in Table 1. Explicit regulation on PSH is in force in 82.3% of the EU member states and in 16.6% of non-EU developed countries, but in less than 30% of the developing countries (Table 1). WV is regulated in three out of four developed countries, but roughly in one out of three developing countries. In many developing countries, OSH legislation includes only prevention of sexual harassment or protection of dignity and religion.

### 3.1. Psychosocial Hazards and WV Regulation in Developed Countries

Most developed countries include some form of regulations on mental health and/or psychological hazards (psychosocial risks, occupational violence or both of them) in their OSH legislation. All of the EU member states available in LEGOSH (*n* = 17) explicitly or implicitly included PSH and WV in their OSH legislation. Most countries in Scandinavia, Continental Europe and the Mediterranean have a specific regulation for this topic. For instance, in Finland, workload factors, lone working, night work and work pauses, as well as harassment and occupational violence are specifically addressed. In Sweden, systematic work environment management includes provisions against the risk of violence or the threat of violence. In Latvia, there is a legal framework for harassment (both personal or through instructions to other people), direct (gender) and indirect discrimination and occupational violence, which can be in the form of physical abuse or sexual harassment. In the Netherlands, the employer is obliged to address psychosocial pressure of work and working conditions policy, which is aimed at preventing sexual intimidation, aggression, discrimination and violence. In France, the law provides protective measures against psychosocial risks and stipulates provisions against moral and sexual harassment. In Italy, all psychosocial hazards, including both psychosocial risk and occupational violence, must be assessed by employers. In Portugal, psychosocial risks including violence, discrimination and sexual harassment are covered under the general duties of the employers. In Spain, there is no data available on psychosocial risks; nevertheless, discrimination and harassment are considered by Spanish law as very serious infringements. In the United Kingdom, psychosocial risks and violence (including verbal abuse and the risk of reasonably foreseeable violence) are covered under general duties, however there is no specific OSH rule. In Ireland, despite the fact there is no data available about psychosocial hazards, employers are required to identify risks of violence at work to implement appropriate safeguards. In Greece, there is a framework agreement to tackle workplace “physical or psychological” violence exercised by co-workers or third parties, though this agreement has not yet been translated into law. In Croatia, any direct or indirect discrimination in the workplace is prohibited, and harassment or sexual harassment is regulated by special legislation, yet there is no explicit regulation for occupational violence. Occupational violence is explicitly included in the rules of prevention by almost all EU member states except for Bulgaria, Romania, and Poland. In Poland, however, the regulation of psychosocial risks includes mobbing that is considered as an unwanted behaviour aimed at or which has the effect of violating someone’s dignity or creating an intimidating, hostile, demeaning or humiliating atmosphere towards an employee.

Developed European countries that are not part of the EU have an approach similar to those in the EU. In Norway, occupational law “foster inclusive working conditions and equality and facilitate adaptations to the employee’s capabilities and circumstances of life”. Moreover, employers must preserve the employees’ integrity and dignity against harassment or other improper conduct (threats, undesirable strain and occupational violence). In Switzerland, law mandates the provision of information on occupational violence, including discrimination and sexual harassment, but not on other PSH.

The legislation of other developed countries outside Europe is less homogeneous than in the EU. Remarkably, we found “no data available” from the USA with regard to PSH. In Canada, the law provides prevention only for WV, not PSH, and in New Zealand, general duty provisions in principle address psychosocial risks and violence, however there are no OSH provisions that explicitly address PSH and WV. In Australia, even if the health of workers includes physical and mental aspects, neither psychosocial risk nor occupational violence are included in its legislation. A synthesis of the legislative framework of developed countries is reported in Table 2.

### 3.2. Psychosocial Hazards and WV Regulation in Developing Countries

The legislation of developing and transition countries is less homogeneous than that of developed countries. In Europe, most of the non-EU developing countries have no specific regulation on this topic. Only Albania and the Former Yugoslav Republic of Macedonia explicitly included PSH in their legislation. In the Russian Federation, OSH legislation explicitly only covers physical health and there is no definition of PSH or WV.

In Dominica and Granada (North America), psychosocial risks and occupational violence are covered by their respective OSH legislation. In Central and South America, PSH is included by only 5 out of 30 countries (16.66%). More specifically, LEGOSH showed “no data available” on this topic for 24 countries. Finally, with regard to WV, out of 10 countries, 33.33% from Latin America consider this topic, and 5 of these countries (50%) focus only on “sexual harassment”. In Colombia, there is a specific resolution on psychosocial risks in the workplace. The termination of the labour contract can be declared both by the employer or employee upon occupational violence. In Venezuela, employers have to ensure a safe and healthy working environment for the full enjoyment of physical and mental faculties of workers, preventing any harassment situations, including physical or psychological violence. In El Salvador, the law provides the institution of psychosocial risks preventive programmes, including training and the participation of an expert in the field and addressing violence against women and sexual harassment in the workplace. In Puerto Rico, the law does not explicitly refer to mental or psychological health, however there is a law indicating a general awareness about psychosocial risks and the importance of preventing them. In Haiti, although the pursuit of workers’ well-being includes “physical, moral, spiritual and material matters”, there is no data about PSH prevention in the workplace. In Saint Lucia, the regulation covers only sexual harassment. In Guatemala, the General Regulation on OSH does still contemplate the need to draft a specific regulation, but, to date, no legislation on the subject has been identified.

In Africa, PSH prevention is mandatory in 14 countries. WV is considered by the legislation of 16 African countries (34.78%), of which 6 exclusively deal with “sexual harassment”. Only 10 countries have OHS regulations in their control of both PSH and WV. In Morocco, the employee may terminate the contract if the employer commits gross insult and incitement of corruption (PSH) or in case the employer commits any form of violence and sexual harassment (WV). In Burkina Faso, the Labour code requires the employer to include initiatives for the prevention of physical and mental violence, including sexual harassment. In Niger, stress is included among the emerging psychosocial risks to be considered by the employer in risk assessment; occupational violence is limited to sexual harassment. In Togo, no employee may be dismissed for having refused the acts of harassment of an employer; moreover, acts of coercion, violence against people and property are prosecuted and punished. According to Comoros’ law, employers must take all necessary measures to prevent mental fatigue of workers, as well as sexual or moral harassment. In Zambia, the employer must adapt the working environment to the employee’s physical, physiological and psychological ability. In Namibia, the law requires employers to consider discomfort, fatigue or psychological stress due to failure to apply ergonomic principles and regulates discrimination and sexual harassment in the workplace. In Uganda, both PSH and WV are regulated. In South Africa, physical assault from the employer, a fellow employee, client or customer is regarded as serious misconduct of the employer, since there is a constitutional obligation to respect and protect the dignity of the employees. Intimidation and sexual harassment are also regulated in Lesotho.

Other African countries only regulate PSH, without specifically considering WV. In Egypt, the employer must ascertain the workers’ fitness from the point of view of their physical, mental and psychological abilities in order to ensure their fitness to work demands, however there is no regulation about WV. In the Central African Republic, regulations include PSH, but not WV. The same regulatory situation exists in other African countries: in Mozambique, Angola and Congo, there are rules for preventing psychosocial risks, but not occupational violence. Conversely, in other countries, the law provides specific rules against violence. In Mauritius, psychosocial risks are not specifically covered by the Labour Act, yet occupational violence, harassment, sexual and verbal abuse, the threat of violence and bullying are explicitly considered. In Tunisia, acts of violence or threats against any fellow worker or person not belonging to the enterprise are regulated. Occupational violence is also regulated in Kenya, Libya and Djibouti. In Rwanda, the law regulates the protection of workers against violence or harassment, however there is no data available on PSH.

In Asia, PSH is a specific topic of OSH in 8 countries, whereas occupational violence is included in the OSH legislation of 10 countries. In China, employers must take measures to protect the physical and mental health of employees and to prevent sexual harassment for female workers in the workplace. In India, psychosocial risks caused by economic pressure are recognized by certain statutes and employers are obliged to minimize instances of sexual harassment in the workplace. In Korea, the employer must evaluate some PSH, such as working for a long time, shift work including night duty, vehicles operation and precision machine control work; sexual harassment by a superior or co-workers is prohibited. In Singapore, despite the law covering both the physical and psychosocial health of persons at work, and WV is regulated, there is no legal provision that specifically addresses PSH in the workplace. The most frequent normative situation is the opposite one, in which PSH is regulated, however there are no laws on violence, or there are laws limited to conduct regarding sexual or religious customs. In the Philippines, the law provides that the workplace be supportive and enhancing of the psychological health of workers, however there is no provision about work-related violence. In Thailand, the legislation covers both physical and psychological health; however, there is currently no specific requirement about WV. Moreover, in China and the Republic of Korea, the concept of WV comprises only “sexual harassment”, whereas, in Vietnam, the law mentions “sexual harassment and violence against domestic workers”. In Qatar, psychological health has not specifically been addressed by OSH legislation. However, the law requires the employer to conduct pre-employment medical examinations to ensure that the worker is psychologically fit for the type of work required. In Oman, there is no rule on PSH, however violence is regulated. In Jordan, there is no data available on PSH; however, WV, i.e., using force, violence, threatening or illegal procedures in assaulting or attempting to violate the right of others, must be prevented. In Saudi Arabia and in Yemen, workplace sexual misconduct is regulated, however there is no legal provision against work-related stress (Table 3).

## 4. Discussion

Our review showed that most countries around the world have not included mandatory PSH risk assessment and WV prevention in their own national OSH legislation. The lack of indications for employers is not justified in light of the scientific evidence that occupational stress causes cardiovascular [23,24] and psychiatric disorders [25,26,27], general health impairment and low levels of well-being [28], and influences the occurrence of injuries [29]. Similarly, WV has significant effects on workers’ health and productivity [30,31,32,33]. We found a difference between developed and developing countries on this topic, showing a higher frequency of legislative measures in developed countries. This undoubtedly shows a shift towards a greater focus on the issues associated with work-related stress and WV by governments and civil societies of some developed countries. However, we found differences among developed countries, also showing that some of these countries fail to take measures that compel employers to prevent psychosocial risks. Moreover, in many countries, WV was prohibited only if it included an attack on sexual or religious customs.

Since the PSH and WV prevention policy may generate some costs for employers, a possible obstacle to its adoption may be of an economic nature. However, beyond the obvious distinction between developed and developing countries, we observed that some of the major world economies in the developing/transition area (China, Russia, India, Brasil) and countries with the highest per capita income (Qatar, Emirates, Singapore) have limited regulation on PSH or WV.

Moreover, it is remarkable that some developed countries do not have national laws that provide for the obligation to evaluate PSH or WV risk in the workplace. In particular, our research was incapable of finding any mandatory regulation in the USA, Australia and New Zealand, while Canada only had regulation for WV. Even if the LEGOSH database does not allow us to exclude that there may be some regional or local regulation, we must conclude that the situation in these countries is different from that of the other developed countries and, in particular, from the EU member states. A confirmation of the fact that these developed countries have no national mandating rules for the assessment of psychosocial risks or violence against workers is provided by training programs that are developed by the national safety and health agencies. In the USA, the Occupational Safety and Health Administration (OSHA) training courses do not include PSH or WV [34]. The opposite is the case for the equivalent EU administration (EU-OSHA) of which their current themes include stress and psychosocial risks [35] and where psychosocial occupational risks in different countries are constantly monitored [36]. The situation is similar for Safe Work Australia (SWA), the Australian government statutory body established in 2008 to develop national policy relating to OHS, which does not include PSH or WV in their model code of practices [37]. The Canadian Centre for Occupational Health and Safety (CCOHS) considers workplace violence and bullying in its health and safety programs, however it does not include stress or other psychosocial risks prevention into the action required under OHS legislation in Canadian jurisdictions [38].

The general framework, therefore, leads us to believe that a distinction among countries does exist, however it could be founded on different cultural and economic models for social development rather than on the availability of economic resources. Our findings show support for the Scandinavian model of workplace health and safety culture, which emerged during the 1970s and inspired the EU one, and is based on a “three-pillar” system of collective bargaining and extensive collaboration involving employers, employees and government [39,40,41], which undoubtedly facilitates the development of policies addressing PSH/WV prevention. On the contrary, many non-EU developed countries, such as the USA, Switzerland and others, leave these important aspects of occupational safety and health to be regulated by market forces rather than institutional actors [42].

With regard to WV, the observed differences may also derive from cultural, religious and socio-political roots. European countries generally recognize both psychosocial risks and occupational violence as important factors according to two European Framework Agreements (2004; 2007) [43,44]. However, the concept of “what” must be protected is substantially different in some non-European countries with respect to European states. For example, in some legislation of developing countries, protection from WV is limited to sexual harassment, consequently ignoring the health risk associated with all other forms of non-sexual violence [45,46,47,48,49].

Our search has the limitations that derive from the data source. Although LEGOSH is the most up-to-date database available, in some cases, the last update ranged between 2013 and 2015. We cannot, therefore, exclude that more recent legislation has been introduced. Moreover, we cannot even rule out the possibility that in some countries legislation exists, but that ILO-LEGOSH researchers have not been able to find it, even if this is unlikely.

The differences we identified among the national laws could roughly be related to differences in the health of workers. Health risk management in the workplace is a complex process that requires hazard identification, risk measurement and the adoption of preventive measures. When all this is not required by law but is entrusted to the will of individual entrepreneurs, this may lead to health problems among workers. Previous studies indeed observed that countries with well-established active labour market policies also have better working conditions and lower levels of work-related stress than unregulated countries [50,51,52,53,54,55]. Inequalities in preventive dispositions may favour inequalities in health between and among workers.

Naturally, the lack of national laws is only the first indication of reduced attention to the prevention of occupational risks. The lack of effective measures for the enforcement of the rules can be another important factor of inequality in the workplace and between workers. A systematic literature review showed that the introduction of regulatory policy levers is often effective in reducing injuries and/or increasing compliance with OHS legislation [56]. The lack of rules for the prevention of psychosocial risk in some developed countries may also reduce the effectiveness of programs for improving occupational health. Studies showed that integrated approaches to promoting and protecting worker health, addressing both environmental and psychosocial factors, are highly effective and responsive to specific productive needs [57,58].

The lack of health and safety laws for the prevention of occupational stress and violence in many areas with a high-income economy is worthy of consideration in relation to the presence of the migration phenomenon, which can greatly aggravate inequalities in workplaces. Migrating workers often have unrecognized mental health needs [53] because migration per se can be a very stress-inducing phenomenon [59]. Migrant workers are often engaged in what is known as 3-D jobs, i.e., dirty, dangerous and demanding [60]. This includes working in an isolated environment, with limited supervision and guidance, which makes them more prone to abuse and exploitation [61]. In cases where social and OSH policies are poor, the distress could lead to mental health consequences or other forms of health complications [62]. The remedy, typical of many authoritarian regimes, of countering or prohibiting migration has turned out to worsen the situation, favouring illegal migration with fewer safeguards [63].

In 2015, the migrant population represented nearly 4% of the total global population aged 15 years and over and cannot be neglected [64]. This leads, furthermore, to the need to homogenise the legislative tools to address psychosocial hazards in all countries. Policymakers should try to fill the existing gaps in national legislation on these topics. Making uniform interventions could also probably facilitate the achievement of the Sustainable Development Goal (SDG 3) established by the United Nations to achieve a better and more sustainable future for all, aiming at a reduction by one third of premature mortality from non-communicable diseases through prevention, treatment and promotion of mental health and wellbeing [65]. SDG 3 provides a rationale and a framework to address mental health from many perspectives with renewed urgency. A needed political action could boost actions to tackle the unequal psychosocial risk assessment policies in the workplace, giving decisive occupational health benefits.

## 5. Conclusions

Our study evidenced marked differences in the legislative framework of work-related psychosocial risks among countries. These differences may lead to unequal levels of worker protection in the workplace, which can be the subject of future research. Further studies should be focused on the relationship between social, economic and cultural factors and safety and health levels in the workplace. The legislation on PSH and WV is more frequently present in developed countries than in developing ones. In many cases, WV is prohibited only if it represents an attack on sexual or religious customs. There are also significant differences within the group of richer countries, where some fail to take measures that force employers to prevent psychosocial risks.

These inequalities in legislation could have adverse effects on global occupational health and health in general. We believe that the present situation is not acceptable, especially in a context of globalization and migrating workforce, thus legislative improvements are needed.

## Figures and Tables

**Table 1 ijerph-16-02470-t001:** Characteristics of occupational safety and health (OSH) legislation about psychosocial hazards and workplace violence in developed and developing countries.

Countries (*n* = 132)	Psychosocial Hazards		Workplace Violence	
	Yes	No/NDA	Yes/Partial *	No/NDA
Developed countries (*n = 23*)				
EU-countries (*n* = 17)	14 (82.3%)	3/0	14 (82.3%)	2/1
Non-EU countries (*n* = 6)	1 (16.6%)	2/3	3 (50%)	2/1
Total (*n* = 23)	15 (65.2%)	5/3	17 (73.9%)	4/2
Developing countries (*n = 109*)				
Europe (*n* = 14)	4 (28.5%)	1/9	2 (14.2%)	1/11
North America (*n* = 2)	0 (-)	0/2	0 (-)	0/2
Central and South America (*n* = 30)	5 (16%)	1/24	5/4 (30%)	0/20
Africa (*n* = 46)	15 (32.6%)	0/31	10/6 (34.7%)	0/30
The Middle East and Asia (*n* = 17)	8 (47%)	1/8	6/4 (58.8%)	0/7
Total (*n* = 109)	32 (29.3%)	3/74	33/15 (44%)	1/70

Note: NDA = No Data Available. * Partial: Including only sexual harassment or protection of dignity and religion.

**Table 2 ijerph-16-02470-t002:** Synthesis of differences among developed countries in Psychosocial Hazards (PSH) and Workplace Violence (WV) regulation.

Type of Regulation	Countries
Explicit provision of PSH and WV in OSH law	Croatia, Cyprus, Finland, France, Greece, Italy, Latvia, Netherlands, Norway, Portugal, Sweden, United Kingdom
Explicit provision of PSH (not WV)	Bulgaria, Poland, Romania
Explicit provision of WV (not PSH)	Canada, Denmark, Ireland, Spain, Switzerland
No provision	Australia, New Zealand, USA

**Table 3 ijerph-16-02470-t003:** Examples of differences in the regulation of workplace violence among developing and transition countries.

Type of Regulation	Countries
Any kind of violence	Albania, Burkina Faso, Colombia, Comoros, India, Jordan, Lebanon, Macedonia, Malaysia, Mauritius, Mexico, Morocco, Namibia, Paraguay, Rwanda, Singapore, South Africa, Togo, Tunisia, Uganda, Uruguay, Venezuela, Yemen, Zambia
Only infringement of sexual or religious custom	Belize, Chile, China, Djibouti, El Salvador, Kenya, Korea, Lesotho, Libya, Niger, Peru, Saint Lucia, Saudi Arabia, Vietnam
No specific provision on WV in OSH law	Angola, Central African Republic, Congo, Egypt, Mozambique, Philippines, Qatar, Russia, Thailand

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
