# Peer review of "Psychosocial Risk Prevention in a Global Occupational Health Perspective. A Descriptive Analysis"

_ijerph, 2019, doi:10.3390/ijerph16142470_

Round 1

Reviewer 1 Report

This study is the first attempt to describe the psychosocial risk regulatory framework in a global environment. 

I have two main reservations. 

First, the author provided a sufficient background of psychosocial hazards in the workplace. However, the rational of this descriptive study is not strong enough for me. It is well known that psychosocial risk prevention is needed, and it is not surprising that developed countries have a higher frequency of legislative measures than developing countries. Therefore, the importance of reviewing the OSH regulatory frameworks globally should be explained in more depth. 

Second, in the results section. the author described some regulatory details for specific countries. However, I found it hard to follow. I would like to suggest the author have a summary table instead of text description. In the table, you could list countries who have similar regulation, who have different and who have no regulation on mental health risk, and the comparison between developed countries and developing countries. I think this would give the reader a much clear view regarding the global perspective on this topic.  

Author Response

We thank the reviewer for the valuable comments, which have helped us to improve our manuscript. 

Comment: This study is the first attempt to describe the psychosocial risk regulatory framework in a global environment. I have two main reservations. First, the author provided a sufficient background of psychosocial hazards in the workplace. However, the rational of this descriptive study is not strong enough for me. It is well known that psychosocial risk prevention is needed, and it is not surprising that developed countries have a higher frequency of legislative measures than developing countries. Therefore, the importance of reviewing the OSH regulatory frameworks globally should be explained in more depth.

Response: As suggested by the reviewer, we have added a discussion about this issue to our background. We thank the reviewer for pointing out that psychosocial risk prevention is needed. We are firmly convinced of this, although perhaps this opinion is not shared by all developed countries. In fact, our study shows that many developed countries have no regulation for stress or violence at work. We believe that this aspect is more important than the difference between developed and developing countries, which could be due to economic and historical reasons. In our revision, we have tried to underline the importance of reviewing the OSH regulatory framework of the different countries (see Lines 89-106).

The revised text is: 

Reviewing the OSH regulatory framework is of great importance because the presence of differences in regulatory provisions could correspond to differences in the levels of safety and health in the workplace. By making these differences explicit in our review, we provide a base for further policy discussion and improvements.

Not all countries in the world utilize the same level of occupational risk prevention. The cultural and social differences that underlie the different regulatory situations are also the basis of different development model. In recent years, the process of globalization and liberalization have often increased health inequities [20]. For this reason, it is relevant to know whether or not the presence of laws or regulations in the countries around the world, really promotes the development of effective OSH services for the prevention of PSH.

In this review, we aimed to evaluate the OSH regulatory frameworks developed by the countries worldwide, with a special focus on the differences between developing and developed countries and within developed countries, which have more resources to achieve occupational risk prevention. More specifically, our study aimed to clarify whether countries around the world require PSH and WV to be assessed by employers through the mandatory occupational risk assessment process. Such a legal requirement could inform policymakers and key stakeholders for improving occupational health and safety policy on PSH and WV at the workplace.

COMMENT: Second, in the results section. the author described some regulatory details for specific countries. However, I found it hard to follow. I would like to suggest the author have a summary table instead of text description. In the table, you could list countries who have similar regulation, who have different and who have no regulation on mental health risk and the comparison between developed countries and developing countries. I think this would give the reader a much clear view regarding the global perspective on this topic.

RESPONSE: As suggested, we have added two summary tables reporting a comparison of countries who have similar regulations in developed/developing countries (see Tables 2-3).

Reviewer 2 Report

This is an interesting paper—it is helpful to understand the range and variation of policies on workplace violence and psychosocial hazards.  To make the paper stronger, I recommend the follow:

·       Where surprising results are found and they are due to a lack of information in the dataset, then the authors should dig deeper. Find another information source to conform that there are actually no policies covering that topic. This is very important for the credibility of the paper.

·       The authors might consider presenting much of their findings in a tabular format. At present, the sections read like long lists. If you create a comparative table, then the text could be focused on a discussion of the differences across and within the table.

·       It was helpful p. 3 to have the mention that criminal laws may also be present. However, reminder is brought in again selectively in 3.2 end of para 1. Instead, a comment should be made at the end of the findings section (applicable to all jurisdictions, not just Russia) that criminal law provisions may exist.

·       At the conclusion of the paper, the authors suggest that harmonization of legislation is needed because of employment migration. However, the argument was made earlier that this legislation is needed because the prevalence of these problems in workplaces. A tidier explanation (or perhaps a clearly layered one) would help to orient the reader.

·       Tidy up some awkward grammatical sections, e.g. Lines 41, 199, 278

Author Response

We are grateful to the reviewer for his/her valuable comments, which helped us to improve the manuscript.

COMMENT: This is an interesting paper—it is helpful to understand the range and variation of policies on workplace violence and psychosocial hazards. To make the paper stronger, I recommend the follow: Where surprising results are found and they are due to a lack of information in the dataset, then the authors should dig deeper. Find another information source to conform that there are actually no policies covering that topic. This is very important for the credibility of the paper.

RESPONSE: We thank the reviewer for the appreciation of our work and for the advice he/she gave us. In the Discussion we have added some references to confirm that what was found using LEGOSH, corresponds to the reality reported by other sources, looking in particular at the national safety administration of developed countries. This is especially true for the unprecedented findings concerning developed countries without laws on violence or stress in the workplace. (see Lines 313-330).

The text added was:

Moreover, it is remarkable that some developed countries do not have national laws that provide for the obligation to evaluate PSH or WV risk at the workplace. In particular, our research was incapable of finding any mandatory regulation in the USA, Australia and New Zealand, while Canada had only regulation for WV. Even if the LEGOSH database does not allow us to exclude that there may be some regional or local regulation, we must conclude that the situation in these countries is different from that of the other developed countries and in particular from the EU member states. A confirmation of the fact that these developed countries have no national mandating rules for the assessment of psychosocial risks or violence against workers is provided by training programs that are developed by the national safety and health agencies. In the USA, the Occupational Safety and Health Administration (OSHA) training courses don't include PSH or WV [34]. The opposite is the case for the equivalent EU administration (EU-OSHA) whose current themes include stress and psychosocial risks [35] and where psychosocial occupational risks in different countries are constantly monitored [36]. The situation is similar for the Safe Work Australia (SWA), The Australian government statutory body established in 2008 to develop national policy relating to OHS, do not include PSH or WV in the model code of practices [37]. The Canadian Centre for Occupational Health and Safety (CCOHS) considers workplace violence and bullying in its health and safety programs, but does not include stress or other psychosocial risks prevention into the action required under OHS legislation in Canadian jurisdictions [38].

C.: The authors might consider presenting much of their findings in a tabular format. At present, the sections read like long lists. If you create a comparative table, then the text could be focused on a discussion of the differences across and within the table.

R.: As suggested, we now try to summarize the results with the use of some summary tables 2 and 3.

C.: It was helpful p. 3 to have the mention that criminal laws may also be present. However, reminder is brought in again selectively in 3.2 end of para 1. Instead, a comment should be made at the end of the findings section (applicable to all jurisdictions, not just Russia) that criminal law provisions may exist.

R.: We have removed the comment related to Russia, because, as we have explained in the methods, we have not considered the criminal laws, present in all the states, but only the laws that concern the work.

C.: At the conclusion of the paper, the authors suggest that harmonization of legislation is needed because of employment migration. However, the argument was made earlier that this legislation is needed because the prevalence of these problems in workplaces. A tidier explanation (or perhaps a clearly layered one) would help to orient the reader.

R.: We completely agree. Indeed, migration is a phenomenon that aggravates existing problems, but the discussion must, first of all, emphasize the importance of the problems existing in the workplace. We have restructured the text in this sense, adding some references. (see Lines 360-368).

The text added was:

Naturally, the lack of national laws is only the first indication of reduced attention to the prevention of occupational risks. The lack of effective measures for the enforcement of the rules can be another important factor of inequality in the workplace and between workers. A systematic literature review showed that the introduction of regulatory policy levers is often effective in reducing injuries and/or increasing compliance with OHS legislation [56]. The lack of rules for the prevention of psychosocial risk in some developed countries may also reduce the effectiveness of programs for improving occupational health. Studies showed that integrated approaches to promoting and protecting worker health, addressing both environmental and psychosocial factors, are highly effective and responsive to specific productive needs [57, 58].

C.: Tidy up some awkward grammatical sections, e.g. Lines 41, 199, 278

R.: As suggested, we have corrected these typos. We have also checked the text for other typos.